# Clusters of Physical Frailty and Cognitive Impairment and Their Associated Comorbidities in Older Primary Care Patients

**DOI:** 10.3390/healthcare9070891

**Published:** 2021-07-15

**Authors:** Sanja Bekić, František Babič, Viera Pavlišková, Ján Paralič, Thomas Wittlinger, Ljiljana Trtica Majnarić

**Affiliations:** 1General Medical Practice, 31000 Osijek, Croatia; sanja.bekic1@gmail.com; 2Faculty of Medicine, University Josip Juraj Strossmayer, 31000 Osijek, Croatia; 3Department of Cybernetics and Artificial Intelligence, Faculty of Electrical Engineering and Informatics, Technical University of Košice, 04201 Košice, Slovakia; viera.pavliskova@tuke.sk (V.P.); jan.paralic@tuke.sk (J.P.); 4Department of Cardiology, Asklepios Hospital, 38642 Goslar, Germany; dr.wittlinger@gmx.de; 5Department of Internal Medicine, Family Medicine and the History of Medicine, Faculty of Medicine, University Josip Juraj Strossmayer, 31000 Osijek, Croatia; ljiljana.majnaric@mefos.hr; 6Department of Public Health, Faculty of Dental Medicine and Health, University Josip Juraj Strossmayer, 31000 Osijek, Croatia

**Keywords:** multimorbidity, primary care, physical frailty, cognitive impairment, latent cluster analysis

## Abstract

(1) Objectives: We aimed to identify clusters of physical frailty and cognitive impairment in a population of older primary care patients and correlate these clusters with their associated comorbidities. (2) Methods: We used a latent class analysis (LCA) as the clustering technique to separate different stages of mild cognitive impairment (MCI) and physical frailty into clusters; the differences were assessed by using a multinomial logistic regression model. (3) Results: Four clusters (latent classes) were identified: (1) highly functional (the mean and SD of the “frailty” test 0.58 ± 0.72 and the Mini-Mental State Examination (MMSE) test 27.42 ± 1.5), (2) cognitive impairment (0.97 ± 0.78 and 21.94 ± 1.95), (3) cognitive frailty (3.48 ± 1.12 and 19.14 ± 2.30), and (4) physical frailty (3.61 ± 0.77 and 24.89 ± 1.81). (4) Discussion: The comorbidity patterns distinguishing the clusters depend on the degree of development of cardiometabolic disorders in combination with advancing age. The physical frailty phenotype is likely to exist separately from the cognitive frailty phenotype and includes common musculoskeletal diseases.

## 1. Introduction

Population aging is a global trend in EU countries [1]. Accompanying this trend is an increase in the number of individuals with multimorbidity (a coexistence of two or more chronic diseases in the same person) and who are showing functional decline, which poses new challenges to healthcare systems, such as high requirements for utilizing healthcare services and long-term care, in particular.

Epidemiologic studies have indicated that multimorbidity increases with age and is associated with a deterioration in mental health and low physical, cognitive, and social functioning [2,3,4]. These observations support our current understanding of the development of common diseases of aging, such as diabetes type 2 (diabetes), cardiovascular disease (CVD), Alzheimer`s dementia, and some types of cancer, as being an integrative part of the aging process [5]. Although, it has been realized that the extent to which these diseases and functional organ impairments are expressed vary between individuals, reflecting interindividual differences in rates of aging, so that the real (biological) age may fall behind or outpace the chronological age [6]. The causes and mediators of such differences are mostly unknown. According to today’s prevailing theory of aging, inflammaging, a variety of stimuli operating at cellular and subcellular levels in the body, contribute to low-grade inflammation as the main driver in the acceleration of aging and the development of age-related diseases [7]. Fat tissue redistribution, which occurs with aging and is clinically visible as the abdominal type of obesity, in particular when it is combined with overnutrition and general overweight/obesity, may substantially contribute to inflammation and metabolic disorders associated with aging and the development of cardiometabolic age-related diseases, such as metabolic syndrome, diabetes, and CVD. Cerebral small-vessel disease, recognized as a pathologic mechanism underlying non-Alzheimer`s cognitive disorders, is considered a part of inflammaging and reflective of the overwhelming influence of metabolic and inflammatory stimuli on pathologic changes in the brain vasculature [8].

An age-related decline in physical and cognitive capabilities can be best described by applying the concepts of physical frailty and mild cognitive impairment (MCI). Both conditions have been proven to independently increase the risk of negative health outcomes, including falls, disability, dementia, hospitalization, institutionalization, and death [9]. Frailty is considered a manifestation of reduced homeostatic reserves in many vital systems that govern neuroendocrine, energy-metabolic, and inflammation-immunologic mechanisms [10]. The transition from a prefrailty to frailty state takes place in parallel with the progression of pathophysiologic disorders, when it becomes increasingly less possible to reverse this syndrome [11,12]. The concept of MCI has been introduced to define a stage of cognitive decline between normal cognition and dementia that can be objectively measured but is still not severe enough to affect the activities of daily living [13]. Although MCI is associated with an increased risk for developing dementia, without additional complementary variables, this measure is not powerful enough to accurately predict dementia [14,15].

Emerging evidence indicates that these two disorders, physical frailty and cognitive impairment, often coexist and mutually interact, thus increasing the risk of each condition for poor health outcomes [16,17]. Although the evidence suggests that these disorders share many risk factors and mechanisms, the knowledge of common pathophysiologic pathways is still low, mainly because these disorders have been studied separately so far as independent entities [18,19].

A new entity, termed cognitive frailty, defined as the coexistence of prefrailty or frailty with MCI, has been established by the international consensus group with the aim to facilitate research on cognitive impairment that is caused by deteriorating physical health, thus distinguishing physical from neurodegenerative causes of cognitive impairment [20].

It is becoming increasingly apparent that the dynamic interplay between chronic diseases and functional impairments, which are modulated by genetic, behavioral, and environmental factors, as well as by applied treatments, directs the rates of age-related decline in physical and cognitive performance [21]. Although prospective epidemiologic studies indicate that physical frailty may be a driver of cognitive impairment, and that the opposite is less likely to occur, our knowledge concerning the exact clustering patterns of physical frailty and cognitive impairment and of their dynamics of change in the aging population is poor [17,22]. There is an increasing expert consensus that screening for cognitive impairment should be performed in all older prefrail and frail individuals with multimorbidity [17,18].

The recent shift in research on multimorbidity from disease counting to disease clustering has revealed disease patterns that could be based on common pathophysiologic pathways [23,24,25]. The aim of the present study was to identify clusters of physical frailty and MCI in a population of older primary care patients and to correlate the identified clusters with comorbidities and chronological age. Differences among clusters in degrees of functional declinemay reflect interindividual differences in rates of aging. Identified clusters will relate these differences to the level of the development of age-related diseases and functional organ impairments, more precisely reflecting the aging process than by using chronologic age alone [6].

## 2. Methods

### 2.1. Study Design and Participants

A cross-sectional study and retrospective analysis of the selected data used from primary care (PC) electronic health records (eHRs) were conducted in 2018 in an academic General Practice (GP) facility in the town of Osijek (currently around 60,000 inhabitants), the administrative center of Eastern Croatia. Due to the poor economic situation in this area, negative demographic trends and population aging have taken place, which has led to a high burden of chronic diseases; higher than this is the average in Croatia.

The analysis included 263 older (≥60 years) ambulatory PC patients who were enrolled at their regular visits or were invited for an interview. According to several rules of thumb, the sample size of 250 participants was sufficiently large to show statistical significance for less complicated latent class analysis (LCA) models, which was the critical analytical method used in this study [26,27,28].

The fact that patients were recruited from one GP facility did not hamper the representativeness of the sample, because older people living in the area have similar living conditions and are generally of a lower socioeconomic status. A good match with the general population was ensured by the fact that, in Croatia, the general population has good access to PC services, and almost all inhabitants are registered on the lists of PC physicians. The data collection from a single practice may even have some advantages by ensuring the uniformity of the diagnostic criteria and terminology that is used in communication with patients and during the diagnostic process. The fact that it was an academic GP facility ensured that the data was collected by a skilled and knowledgeable PC physician, which could guarantee the high level of data accuracy.

Of approximately 2000 patients registered in this GP facility, about a quarter were older individuals, and about a half of them entered the study. We used for analysis only community-dwelling patients to whom preventive measures, if applied, may still be beneficial and not those in home care programs or in institutions. The exclusion criteria were also acute medical conditions, exacerbations of chronic conditions, and diagnoses of psychosis or dementia. Excluded from the study were also several patients with incomplete health records. We already have two papers published using the same dataset. In the first published paper, an unsupervised learning algorithm, k-means, was applied on the data obtained from 159 patients who were enrolled first to identify clusters of numerical variables indicating mental disorders, cognitive impairment, physical frailty, and laboratory tests [29]. Information on diagnoses of chronic diseases and some functional and sensory organ impairments was used to complement the description of these clusters. When the data collection was finished, we applied the supervised latent class analysis (LCA) model on the full-sized sample of 263 patients to identify individuals with different stages of cognitive impairment and physical frailty who showed a tendency to cluster together [30]. In that paper, we presented the first part of the complex analysis, where we assessed how membership in a cluster is influenced by performances on tests of mental disorders, anxiety, and depression and by specific cognitive test tasks. In this paper, we presented the second part of the analysis, where we analyzed the differences among clusters concerning comorbidities and functional/sensory organ impairments.

### 2.2. Data Collection

The selection of variables that were used for analysis was based on knowledge and data availability. Data were collected from eHRs on the number and types of diagnoses of chronic diseases, the total number of prescribed medications and the number of medications with an effect on mental functions, and on laboratory tests that are routinely performed in PC to check patients` health status and which indicate metabolic disturbances and the status of inflammation and nutrition. Diagnoses of chronic diseases were recorded according to the international disease coding system (ICD-10). The laboratory test results were used from chronic disease surveillance programs and preventive check-ups and were not older than a year. The systematic way of data recording in these platforms has ensured a high level of data completeness. Only in a few cases was data missing, indicating the C-reactive protein or glomerular filtration rate, and these patients were excluded from the analysis. The laboratory test results were assessed according to appropriateness for participants’ age and health status by comparing them with the laboratory reference values and recommendations from the international guidelines for managing common chronic conditions [31,32]. Information on functional/sensory organ impairments was gathered from eHRs and by patient interviews. Anthropometric measures, BMI (body mass index), a measure of general nutrition, waist circumference (a measure of abdominal obesity), and the mid-arm circumference (a measure of muscle mass loss) were performed during patient visits to add to the information on the nutritional and health status of participants who were recruited to the study [33].

To determine the level of physical frailty of participants, we used the Fried phenotypic model, which is the best-validated of available, similar measures [34]. Based on five criteria, weight loss, slow walking speed, weak grip strength (measured by the handgrip dynamometer), a subjective feeling of exhaustion, and reduced activity, this model indicates whether an individual is prefrail (1 to 2 positive criteria), frail (≥3 positive criteria), or robust (no one positive criterion).

For measuring MCI, as a component of the cognitive frailty phenotype, the international consensus group recommended the Clinical Dementia Rating Scale [20]. To screen participants for MCI in this study, we used the Mini-Mental State Examination (MMSE) test, which has been broadly validated also in the elderly Croatian population [35]. This test consists of several domains, indicating either memory-related or non-memory-related (executive) functions. The MMSE cut-offs were adjusted for the participants’ level of education based on the MMSE cut-off values for the Croatian population. This cut-off was 24/25 (of the maximum 30) for screening among older individuals in the general population and 26/27 for screening among those with a higher level of education (defined as ≥14 years of education). The MMSE test is more sensitive for diagnosing severe cognitive impairment (scores ≤ 17) than for distinguishing between cognitively healthy individuals and those with MCI and cannot distinguish between different types of dementia (Alzheimer’s type vs. vascular type).

Information on the sociodemographic characteristics and medical history of the participants are presented in Appendix A. The numerical variables are presented as the mean and the standard deviation (SD) or as the median and the interquartile range (Appendix A). The categorical variables are presented with the absolute numbers and frequencies (%) (Appendix A).

### 2.3. Statistical Analysis

The LCA method was used to identify subgroups, latent classes, as statistically distinct and clinically meaningful patterns that optimally comprehend the heterogeneity of participants in the sample regarding their achievements on the MMSE test and the Fried frailty score [30,36].

Differences in distributions of numerical variables among the clusters were analyzed using the one-way analysis of variance (ANOVA) or Kruskal–Wallis rank sum test, depending on whether numerical variables showed a normal distribution. This analysis was followed by the Games-Howell post hoc test. Differences in the categorical variables were assessed using the chi-square (χ2) test and Fisher`s exact test, where appropriate. Bar diagrams were used to visualize the distributions of those categorical variables for which differences among the clusters reached statistical significance.

To assess how the examined variables are associated with membership in a cluster, we used a multinomial logistic regression (MLR) model from R statistics. A cluster consisting of individuals with the best cognitive and physical performances was used as a control. We analyzed the impact of age and gender on clusters` membership in a separate MLR model. Four other models were created to show the impact on clusters` membership of (1) the level of comorbidity, presented with variables indicating the number of comorbidities and functional/sensory organ impairments and the number of prescribed medications and medications with an effect on mental functions, (2) the health-related status, presented with variables indicating anthropometric measures and laboratory tests, (3) particular diagnoses of chronic diseases, and (4) functional/sensory organ impairments. Before generating these models, we checked all numerical variables in the input on collinearity, using a simple linear correlation analysis, and on multicollinearity, using the variance inflation factor (VIF) as an indicator. Variables with a high level of collinearity were not included in the models. In the third and the fourth models, only variables that were shown significant in the analysis of the differences entered the model. The AIC (Akaike Information Criterion) was used to measure the quality of the model’s predictive performance [37].

## 3. Results

Members of the first cluster showed better cognitive and physical performance than members of the other three clusters. This cluster was therefore termed highly functional (HF). In members of the second cluster, the performance on the MMSE test was decreased, but the physical performance was good (low average frailty score); the cluster was therefore termed cognitive impairment (CI). The third cluster was termed cognitive frailty (CF), as members of this cluster showed low physical performance (increased average frailty score) and low cognitive performance (decreased average score on the MMSE test). Finally, the fourth cluster was termed physical frailty (PhyF), as its members had low physical performance (similar to members of the CF cluster) but well-preserved cognitive performance (Table 1).

Table 1 also shows that individuals in the HF cluster are younger than those in other clusters, whereas those in the CF cluster are significantly older than those in CI. Individuals in the PhyF cluster are older than those in the CI cluster and younger than those in the CF cluster, but the differences are not significant. There were no significant differences in the distributions by gender (M:F) within the clusters, except for the PhyF cluster, in which women were dominant (17:1) (χ2 test, *p* < 0.05).

Table 2 shows that the PhyF cluster contains only frail individuals and that the CF cluster contains a high proportion of frail individuals and a smaller proportion of prefrail individuals. In contrast to these clusters, in the CI and HF clusters, a prevalent proportion of the individuals is prefrail (44.1% and 68.3%, respectively), and none are frail.

The majority of individuals in clusters characterized by decreased cognitive function had MCI (74.6% in the CI cluster and 90.5% in the CF cluster, respectively), whereas this proportion was much smaller in the other two clusters (11.1% in the HF cluster vs. 38.9% in the PhyF cluster).

Table 3 shows that clusters in which frailty individuals are dominant (the PhyF and CF clusters), compared to clusters in which individuals are at the stage of prefrailty (the HF and CI clusters), have a higher number of chronic disease diagnoses and prescribed medications, including medications affecting mental functions. The highest number of individuals with functional/sensory organ disorders are allocated to the cluster representing the physical frailty phenotype (the PhyF cluster).

Individuals in the CF cluster, in whom the cognitive frailty phenotype is dominant, had the lowest values for the variables indicating mid-arm circumference, HDL cholesterol, hemoglobin, erythrocyte count, and glomerular filtration rate (a marker of renal function) (Table 4).

It can be seen in Table 5 that the diagnoses of chronic diseases with the most impact for distinguishing comorbidity profiles among the clusters include chronic heart disease, coronary artery disease, upper gastrointestinal tract disorders, osteoporosis, osteoarthritis, low back pain, and anxiety/depression. Of the functional/sensory organ disorders, significant differences among the clusters were shown for falls, walking difficulties, and chronic pain. The proportion of individuals with three or more diagnoses of chronic diseases (indicating the status multimorbidity) in particular clusters is as follows: 55.9% (HF), 47.6% (CI), 78.9% (CF), and 88.9% (PhyF).

Figure 1 shows that participants with the diagnosis of chronic heart disease are mostly allocated to the third (CF) cluster, then to the second (CI) cluster and the fourth (PhyF) cluster. Participants diagnosed with coronary artery disease are mostly allocated to cluster 3 (CF) and then to cluster 4 (PhyF). The diagnosis of upper gastrointestinal tract disorders is more prevalent in clusters that are marked by physical frailty (the CF and PhyF clusters) than in the other two clusters (the HF and CI clusters). In fact, upper gastrointestinal tract disorders are mostly present in cluster 4 (PhyF). The diagnoses of osteoporosis and low back pain (syndroma lumbale) are mostly present in cluster 4 (PhyF), whereas the diagnosis of osteoarthritis is more prevalent in clusters CF and PhyF than in the other two clusters (HF and CI). The frequency of the diagnosis of anxiety/depression is relatively high in all clusters, but the highest frequency for this diagnosis is found in cluster 4 (PhyF).

Most participants who experienced falls were allocated to clusters 3 (CF) and 4 (PhyF). Those in cluster 4 (PhyF), more often than those in the cluster 3 (CF), experienced falls with bone fractures. Subjective walking difficulties were expressed mostly by participants in clusters 3 (CF) and 4 (PhyF). Chronic pain was a hallmark of cluster 4 (PhyF).

The MLR model presented in Table 6 shows that increased age had an impact on memberships to all three pathologic clusters (CI, CF, and PhyF). The gender imbalance only had an impact on the PhyF cluster.

As shown in Table 7, Table 8, Table 9 and Table 10, the variables that best characterize cluster 2 (CI) as compared to the control cluster 1 (HF) included increased fasting blood glucose, increased hemoglobin, and the diagnosis of chronic heart disease.

Having decreased values for HDL cholesterol, mid-arm circumference, and glomerular filtration rate (a measure of decreased renal function), together with walking difficulties and the diagnosis of chronic heart disease, increased the probability of belonging to cluster 3 (CF).

The most prominent clinical characteristics of cluster 4 (PhyF) included decreased renal function, as indicated by the variable glomerular filtration rate, and the highest rate of functional/sensory organ impairments—in particular, including chronic pain, walking difficulties, and falls. Of comorbidities specifically associated with cluster 4 (PhyF) were diagnoses of osteoporosis and anxious–depressive disorders.

## 4. Discussion

The identified clusters (latent classes) represent patterns of two main age-related functional disorders, physical frailty and cognitive impairment, that most optimally describe the functional heterogeneity of older, ambulatory PC patients. Indeed, trajectories for rates of aging have not yet been identified; therefore, dividing an older population into such clusters can show individuals who share similar levels of risk for some negative health outcomes [5]. An assessment of the possible at-risk individuals in the clusters were described by many sociodemographic and health-related characteristics. As there is no adequate research framework for investigating multimorbidity, this method can be applied to manage older patients with multimorbidity in a more integral manner than is currently possible when chronic diseases are considered as independent entities [38,39].

Overall, it can be said that HF and CI clusters represent the early stages of frailty (all individuals are robust or prefrail) and CF and PhyF clusters, in which cognitive frailty and physical frailty phenotypes are dominant and represent the final pathways in the development of frailty. Accordingly, individuals in the two latter clusters are generally older, present with more chronic conditions, and use more medications than individuals in the two former clusters. These results support evidence suggesting that the accumulation of comorbidities with age, together with the effect of polypharmacy that accompanies it, governs the transitions of an individual’s health status to states of greater frailty and disability [22,40,41]. It is important to know how older persons in a population are distributed into these clusters, because only when multimorbidity is combined with frailty does it significantly increase the vulnerability of older persons for different stressors, predisposing them to increased mortality [41].

Differences between the CI and CF clusters in the rates of cognitive performance, together with the switch in participation from the dominant participation of prefrail to the dominant participation of frail individuals, between these two clusters support the evidence indicating that cognitive performance progressively declines across the frailty states and that, in prefrail individuals, cognitive impairment is an early sign of comorbidity-related cerebral involvement [22]. Moreover, this result supports the knowledge indicating that the likelihood of adopting the cognitive frailty phenotype strongly depends on an advancement in age [20].

While the effect of age is obviously important for functional decline to develop, the effect of pre-existing health conditions and behavioral coping strategies may direct the course of the pathophysiology disorders, either towards the development of the physical frailty phenotype or the cognitive frailty phenotype. A better understanding of these external influences could improve our capabilities to cope with the modifiable factors that accelerate aging. To reveal if there is a well-functioning group among very old (80+) individuals corresponding with the course of aging, termed as successful aging, only a large-scale study could give an answer [5]. These statements can better come to one`s senses if analyzing clusters separately from each other or in comparison to each other. Thus, individuals in the HF cluster are the youngest and healthiest. However, they are not wholly free from chronic medical conditions. The moderate presence of prefrail individuals in this cluster (44.1%) can be considered as what Strandberg called “primary frailty” (a vicious cycle in which mild frailty precedes and potentiates the development of most comorbidities) [42]. This stage of frailty, in individuals in the HF cluster, can be explained by increased BMI and waist circumference values, indicating overweightness in combination with the abdominal type of obesity. Important to know in these terms is that the global pandemic of obesity, also affecting the older part of the population, can modify the expression of frailty, which was originally viewed as a state of the body shrinking [34,43]. The abdominal type of obesity represents a sign of ectopic fat accumulation and is detrimental to the development of age-related diseases by contributing to increased systemic inflammation [7]. In muscles, this ectopic fat storage is associated with muscle wasting and weakness, reducing the physical performance in obese individuals [43]. Obesity can further contribute to the development of frailty by acting through obesity-related comorbidities, such as anxious–depressive disorders and chronic lumbar pain, as also indicated by our results, by mechanisms such as reduced mobility and motivation for activities [44].

A comparison of the HF and CI clusters has shown that these clusters share many clinical characteristics, such as the number of chronic diseases and prescribed medications, fairly justified anthropometric indices of obesity, and relatively good renal function. It is expected, as these two clusters represent the early stages of frailty. Yet, they differ from each other in that the individuals in the CI cluster are significantly older and have worse CV profiles, the characteristics of which include higher rates of diabetes and chronic heart disease, longer diabetes duration, and worse diabetes control, as indicated by higher fasting serum glucose (chronic hyperglycemia). Based on its worse cardiometabolic profile, the CI cluster exhibits higher rates of prefrail individuals than the HF cluster (68.3% vs. 44.1%). It is due to the fact that both diabetes and CVD are considered a part of inflammaging and are also closely associated with frailty [45,46]. By putting these results into a broader context of inflammaging, then a wide range of comorbidities with a common pathophysiologic background, that may precede or overlap with CVD, also contribute to the close association of CVD with frailty [47,48].

According to the inflammaging theory, metabolic and inflammatory factors, by acting over time in a vicious cycle, may intensify cardiometabolic comorbidities [5,7]. Differences between the CI cluster and the HF cluster in expressing cognitive impairment (74.6% vs. 11.1% of the members with MCI and a decreased average MMSE score in the CI cluster but not in the HF cluster) can also be viewed in this context. In this case, the cerebral small-vessel disease is thought to be that structural correlate of the brain that makes a link between the intensification of cardiometabolic disorders and worsening of cognitive function [8]. 

Further, in the same context, our results indicate that individuals in the CF cluster, who are significantly older than those in the CI cluster, also have more CVD. This worse CV profile can explain the higher frailty rate and worse MMSE score of individuals in the CF cluster. A prominent feature of this profile is the markedly decreased renal function, which, in the CF cluster but not in the CI cluster, reached the level of chronic kidney disease (glomerular filtration rates <60 mL/min/1.73 m2) [31]. Impaired renal function is a common and concomitant disorder of CVD and cardiometabolic conditions and associated with increased inflammation and the risk of developing malnutrition, sarcopenia (muscle wasting), and frailty [49,50]. All these conditions were found to overlap in the geriatric population, and sarcopenia is increasingly being considered a marker of frailty [51,52].

What else matters when considering the effect of chronic renal impairment on developing the cognitive frailty phenotype, as our results and evidence indicate is the level of this impairment and of the burden of associated disorders [53]. In this regard, individuals in the CF cluster have significantly lower renal function and a higher burden of CVD than individuals in the CI cluster and are also characterized with higher levels of inflammation/malnutrition, as indicated with low HDL cholesterol and a higher level of muscle loss (sarcopenia), as indicated by the lower mid-arm circumference [33,54]. This pathophysiologic background, associated with inflammation–malnutrition and sarcopenia, may underlie the fully developed frailty state in individuals in the CF cluster.

As underscored by our results, individuals in the CF cluster do not differ significantly from those in the PhyF cluster in age, the level of comorbidity and medicalization, and the degree of renal function decline, but they are, nevertheless, characterized by significant cognitive impairment, while individuals in the cluster PhyF are not. Therefore, chronic kidney disease must coexist with some higher degree of CVD expression for cognitive impairment to occur. This conclusion arises from the results indicating that there is a difference between these two clusters in the level of the expression of CVD (including diagnoses of chronic heart disease and coronary artery disease), this level being higher in the CF than in the HF cluster. At higher levels of CV comorbidities, we would expect that the level of inflammation also increases, governing the development of clinically significant malnutrition and muscle loss [55]. According to the inflammaging theory, accelerated cerebral small-vessel disease is a result of the action of intensive cardiometabolic factors and increased inflammation on the cerebral vasculature or of the long duration of these factors [8].

We could not show that there are variations in the levels of inflammation among the clusters, but the real reason could be the limited scope of the laboratory tests used in the study. It is becoming increasingly clear that the commonly used inflammatory marker, CRP, also used in this study, is not suitable for all clinical situations and that only a set of variables, indicating related and overlapping disorders, would be effective for detecting variations in the levels of inflammation in older population groups [56]. In case our hypothesis is true, the total burden of cardiometabolic disorders and the level of inflammation/malnutrition and muscle loss would be a better correlate of decreased cognitive function and the presence of a cognitive frailty phenotype than just decreased renal function. Evidence that the coexistence of kidney and heart diseases, relative to the stages of progression of these disorders, contributes to the development of malnutrition, inflammation, frailty, and cognitive impairment is scarce, however, since these disorders have only been examined separately as two independent disorders [57,58].

What else would be important, is an interplay between different disorders and the dynamics of progression, which only could be assessed by longitudinal examinations. According to some observations, if frailty develops before cognitive impairment, dementia will not develop, in contrast to what happens when cognitive impairment continues to progress in parallel with an advancement in the frailty status [59].

This is likely to be indicated, although indirectly, by our results showing that a disease pattern that specifically marks the PhyF cluster and that can be used to help explain the physical frailty phenotype, a hallmark of this cluster, is markedly different from the comorbidity patterns of clusters that are characterized with significant cognitive impairment (the CI and CF clusters). A disease pattern that typically marks the PhyF cluster includes the highest expression of functional/sensory organ impairments, especially concerning those that are known to accompany musculoskeletal diseases, such as walking difficulties, chronic pain, and falls; common musculoskeletal diseases (in particular, osteoporosis and lower back pain); and mental disorders (anxiety and depression). In addition, only in this cluster does gender imbalance play a significant role in cluster membership, with women being dominant. 

Like our findings, other evidence also suggests that musculoskeletal diseases are more prevalent in women than in men and, in particular, in older women with multimorbidity [60]. Musculoskeletal diseases have been recognized as a leading cause of physical disability and associated with chronic pain and frailty [61,62]. When integrating several pieces of this evidence, there is indication of a close association between chronic pain and mental disorders, anxiety, and depression and that mental disorders usually coexist with musculoskeletal diseases in comorbidity patterns [60,63]. Older persons with chronic pain experience and anxiety and depression use opioid analgesics or psychotropic medications more often than others [64]. These medications might contribute to the higher expression of functional organ impairments and the development of the physical frailty phenotype in individuals in the PhyF cluster [65].

Although individuals in the CF and PhyF clusters share similar levels of comorbidity of chronic diseases and medications prescription and, in particular, the diagnosis of osteoarthritis was recorded at higher rates in these clusters than in clusters where frailty is in early stages (the HF and CI clusters), which could contribute to higher levels of inflammation in the CF and PhyF clusters, the hallmark of the PhyF cluster, as indicated by our results, includes a combination of musculoskeletal disorders—in particular, osteoporosis and low back pain syndrome, with anxiety and depression—associated with the domination of female gender [66]. In this regard, the description of the clinical profile of individuals in the PhyF cluster integrates several pieces of evidence, indicating that women are more prone than men to anxiety/depression, multimorbidity, musculoskeletal diseases, and frailty [62,67,68]. There are opinions that a higher psychological vulnerability is the common proxy for developing musculoskeletal diseases and frailty in older women, with inadequate coping mechanisms and obesity having mediating roles [68].

As indicated by our results, objectively visible or subjectively experienced walking difficulties may serve as a simple sign for recognizing older, frail individuals, regardless of their cognitive function status. 

## 5. Strengths and Limitations

In this study, we presented an innovative approach of how to select older people in the population based on the levels of physical frailty and cognitive impairment, considered together, as clusters, which approach, to some extent, reflects differences in the rates of aging and could be important from a prognostic perspective. However, this study also had some limitations that did not allow generalization of the results. These limitations included the low number of participants, especially the small number of individuals in the clusters indicating physical frailty and cognitive frailty phenotypes, which represent the final pathways in the development of frailty. The small size of these clusters may partly be a consequence of the participants` recruitment bias, due to the fact that many frail and immobile persons were not covered by the study. In addition, very old persons (old 80 years and more) were mostly uninvolved, which may have impaired the real picture of the distribution of functional impairments among older people in the community. Further limitations included a low sensitivity of the applied MMSE test for detecting early signs of cognitive impairment and the lack of some variables indicating inflammation, because they were not recorded in the electronic health records.

## 6. Conclusions

This study aimed to identify clusters of physical frailty and cognitive impairment in a population of older (≥60), ambulatory, primary care patients. A comorbidity pattern that may distinguish the clusters depends on the degree of development of cardiometabolic disorders in combination with advancing age. The physical frailty phenotype is likely to exist separately from the cognitive frailty phenotype. A distinction between the two is likely to be related to variations in the expression of CVD and musculoskeletal diseases and to a gender-related predisposition for chronic diseases.

## Figures and Tables

**Figure 1 healthcare-09-00891-f001:**
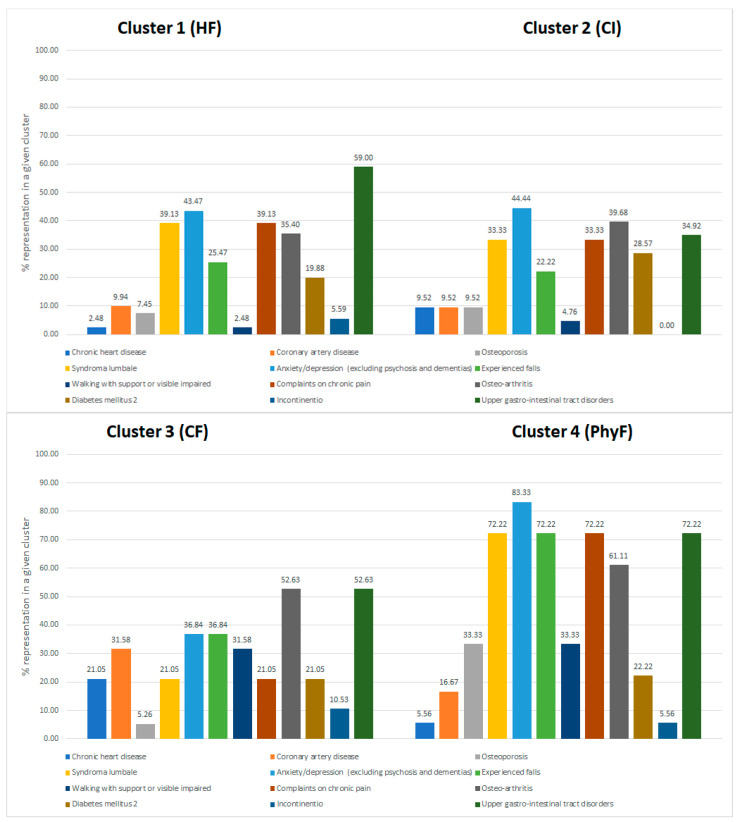
Graphical presentation of the differences among clusters in the diagnoses of chronic diseases and functional/sensory organ impairments.

**Table 1 healthcare-09-00891-t001:** Average scores on the frailty and MMSE tests and average age across the clusters. Division of members of the clusters according to gender.

	Cluster	Number of Patients (M:F)	Average Score ± SD *	*p*-Value (Post-Hoc)	Age (Year) Average ± SD	*p*-Value (Post-Hoc)
**Frailty**	HF	161 (62:99)	0.58 (0.721)	<0.001 HF < CI, CF, PhyF	69.40 (5.455)	<0.001 HF < CI, CF, PhyF
	CI	63 (22:41)	0.97 (0.782)	<0.001 CI < CF, PhyF	72.33 (6.611)	<0.001 CI > HF < CF
	CF	21 (6:15)	3.48 (1.123)	<0.001 CF > CI, HF	78.62 (5.792)	<0.001 CF > CI, HF
	PhyF	18 (1:17)	3.61 (0.777)	<0.001 PhyF > CI, HF	74.72 (6.515)	<0.001 PhyF > HF
	Total	263	1.11 (1.286)		71.20 (6.434)	
**MMSE**	HF	161	27.42 (1.556)	<0.001 HF > CI, CF, PhyF		
	CI	63	21.94 (1.958)	<0.001 CI > CF < PhyF		
	CF	21	19.14 (2.308)	<0.001 CIF > CI, HF, PhyF		
	PhyF	18	24.89 (1.811)	<0.001 PhyF > CI, CF < HF		
	Total	263	25.27 (3.398)			

Note: HF: highly functional, CI: cognitive impairment, CF: cognitive frailty, and PhyF: physical frailty. * Higher scores on the frailty test indicate a higher level of physical frailty, whereas higher scores on the MMSE test indicate a higher level of cognitive function. The results of the post hoc test are represented by a formulation like the cluster combinations HF < CI HF < CF and HF < PhyF CI are significantly different from each other.

**Table 2 healthcare-09-00891-t002:** Division of members of the clusters according to their frailty status and MCI diagnosis.

		Within Clusters				*p*-Value *	All
		HF	CI	CF	PhyF		
Prefrail	N	71	43	3	0	<0.001	117
	% within a cluster	44.1%	68.3%	14.3%	0.0%		44.5%
Frail	N	1	0	18	18	<0.001	37
	% within a cluster	0.6%	0.0%	85.7%	100.0%		14.1%
MCI **	N	18	47	19	7	<0.001	91
	% within a cluster	11.1%	74.6%	90.5%	38.9%		34.6%
All		161 100.0%	63 100.0%	21 100.0%	18 100.0%		263 100.0%

Note: HF: highly functional, CI: cognitive impairment, CF: cognitive frailty, and PhyF: physical frailty. * Pearson chi-square or Fisher’s Exact test, where appropriate. ** MMSE cut-offs for mild cognitive impairment (MCI) adjusted for level of education; levels in the Croatian population ≥65 years were set at ≤24 for education level <14 years and at ≤26 for education level ≥14 years.

**Table 3 healthcare-09-00891-t003:** Differences among individuals in the clusters in the level of comorbidity.

Variable	Median (Interquartile Range) Mean ± SD *				*p*-Value **	Games-Howell Post Hoc Test
	HF	CI	CF	PhyF		
Total number of diagnoses	3.00 (2.00)	3.00 (2.00)	3.84 (2.19) *	4.67 (1.88) *	0.0006	PhyF > HF PhyF > CI
Total number of prescribed medications	3.00 (3.00)	3.00 (3.00)	4.10 (1.97) *	5.17 (2.12) *	0.005	PhyF > HF PhyF > CI
Total number of medications with effect on mental functions	3.00 (2.00)	2.00 (2.00)	3.10 (1.45) *	4.17 (1.62) *	**0.01**	PhyF > HF PhyF > CI
Total number of sensory/functional disorders	2.00 (1.00)	1.00 (1.00)	2.00 (1.05) *	3.00 (1.00)	**0.009**	PhyF > HF PhyF > CI PhyF > CF

Note: *p*-values shown in bold are significant (significance level = 0.05). *—values of mean ± SD were used when Shapiro-Wilk’s test confirmed the normality, **—non-parametric Kruskal-Wallis rank sum test was applied. The results of the GW post hoc test are represented by a formulation like the cluster combinations PhyF > HF and PhyF > CI are significantly different from each other.

**Table 4 healthcare-09-00891-t004:** Differences among individuals in the clusters in the health status described with anthropometric measures and laboratory tests.

Variable	Median (Interquartile Range) Mean ± SD *				*p*-Value **	Games-Howell Post Hoc Test
	HF	CI	CF	PhyF		
BMI (kg/m^2^)	29.73 (5.58)	30.35 (4.45) *	28.13 (4.83)*	28.53 (4.85)	0.19	
Waist circumference (cm)	99.0 (16.00)	101.70 (11.47)*	94.53 (11.18) *	96.61 (16.86)*	0.12	
Mid-arm circumference (cm)	32.00 (3.00)	31.59 (3.63) *	28.79 (3.12) *	30.25 (2.75)	**0.003**	CF < HF CF < CI
Fasting glucose (mmol/L)	5.50 (1.60)	5.90 (1.65)	5.30 (1.20)	5.70 (1.95)	0.18	
Total cholesterol (mmol/L)	5.76 (1.35) *	5.75 (1.24) *	5.93 (1.37) *	6.23 (1.53) *	0.21 **	
LDL cholesterol (mmol/L)	3.60 (1.40)	3.46 (1.06) *	3.46 (1.06) *	3.93 (1.32) *	0.55	
HDL cholesterol (mmol/L)	1.40 (0.40)	1.30 (0.45)	1.22 (0.31) *	1.59 (0.37) *	**0.01**	PhyF > CF
Triglycerides (mmol/L)	1.70 (0.90)	1.80 (0.95)	1.50 (0.55)	1.40 (0.60)	0.10	
Glomerular filtration rate (mL/min/1,73 m^2^)	90.43 (25.50) *	86.08 (29.07) *	67.53 (20.69) *	71.72 (24.08) *	**0.0001** **	CF < HF CF < CI PhyF < HF
C-reactive protein (mg/L)	2.20 (3.20)	2.20 (3.25)	2.40 (4.85)	1.60 (2.10)	0.94	
Hemoglobin (g/L)	138.00 (15.00)	137.00 (15.00)	130.60 (12.11) *	132.20 (22.80) *	0.03	CF < HF
Erythrocyte count (x 10^12^/L)	4.62 (0.43) *	4.56 (0.34) *	4.33 (0.40) *	4.62 (0.42)	0.03	CF < HF

Note: *p*-values shown in bold are significant (significance level = 0.05). *—values of mean ± SD were used when Shapiro-Wilk’s test confirmed the normality, **—non-parametric Kruskal-Wallis rank sum test was applied. The results of the GW post hoc test are represented by a formulation like the cluster combinations CF < HF and CF < CI are significantly different from each other.

**Table 5 healthcare-09-00891-t005:** Differences among individuals in the clusters in particular diagnoses of chronic diseases and functional/sensory organ impairments.

Diagnosis	*p*-Value	Diagnosis	*p*-Value
Hypertension	0.14	Osteoporosis (confirmed)	**0.005**
Diabetes mellitus type 2	0.078	Severe osteoarthritis	0.078
Chronic obstructive pulmonary disease	1.00 *	Low back pain	**0.008**
Asthma or allergic rhinitis	0.575	Parkinson`s disease	0.384 *
Chronic heart disease (failure)	**0.005** *	Urogenital diseases	0.178
Coronary artery disease	**0.039**	The thyroid gland dysfunctions	0.317
Cerebrovascular disease	0.401	Anxiety/depression	**0.011**
Periphery artery disease	0.052 *	Incontinent and other urinary bladder disorders	0.078 *
Upper gastrointestinal tract disorders	**0.030**	Significant visual loss	0.378
Chronic hepatic disorders	1.00 *	Registered hearing impairment or communication difficulties due to hearing loss	0.116
Malignant disease	0.203	Experienced falls	**<0.001**
Chronic pain complaints	**0.008**	Walking with support or visible impaired	**<0.001**
		≥3 Dg of chronic diseases	**<0.001**

Note: Fisher’s exact test (marked with *, when participants in a cluster < 5% or *N* ≤ 10). *p*-values shown in bold are significant (significance level = 0.05).

**Table 6 healthcare-09-00891-t006:** A multinomial logistic regression model indicating differences in the clinical profiles of individuals in the clusters according to their age and gender.

	Cluster CI		Cluster CF		Cluster PhyF	
	z-Value	OR	z-Value	OR	z-Value	OR
Gender = male					−2.29	0.17 (0.05–0.60)
Age	3.46	1.09 (1.05–1.15)	4.96	1.29 (1.19–1.40)	3.75	1.19 (1.10–1.28)

**Table 7 healthcare-09-00891-t007:** A multinomial logistic regression model indicating the differences in the levels of comorbidities among individuals in the clusters.

	Cluster CI		Cluster CF		Cluster PhyF	
	z-Value	OR	z-Value	OR	z-Value	OR
Total number of sensory/functional disorders			−1.92	0.72 (0.64–1.51)	2.92	2.34 (1.45–3.78)

**Table 8 healthcare-09-00891-t008:** A multinomial logistic regression model indicating the differences in their health status, defined by anthropometric measures and laboratory tests, among individuals in the clusters.

	Cluster CI		Cluster CF		Cluster PhyF	
	z-Value	OR	z-Value	OR	z-Value	OR
HDL cholesterol			−2.11	0.12 (0.02–0.63)		
Fasting glucose	2.64	1.23 (1.10–1.40)				
Mid arm circumference			−2.15	0.84 (0.73–0.96)		
Glomerular filtration rate			−2.35	0.97 (0.96–0.99)	−2.37	0.97 (0.95–0.99)
Haemoglobin	3.04	1.06 (1.03–1.09)				

**Table 9 healthcare-09-00891-t009:** A multinomial logistic regression model indicating the differences in their health status, defined by particular diagnoses of chronic diseases, among individuals in the clusters.

	Cluster CI		Cluster CF		Cluster PhyF	
	z-Value	OR	z-Value	OR	z-Value	OR
Chronic heart disease (failure) = yes	2.20	4.78 (1.49–15.35)	1.89	5.05 (1.23–20.77)		
Dg of osteoporosis (confirmed) = yes					2.24	4.30 (1.47–12.54)
Dg of anxiety/depression (excluding psychosis and dementias) = yes					2.10	4.17 (1.36–12.73)

**Table 10 healthcare-09-00891-t010:** A multinomial logistic regression model indicating the differences in their health status, defined by particular functional/sensory organ impairments, among individuals in the clusters.

	Cluster CI		Cluster CF		Cluster PhyF	
	z-Value	OR	z-Value	OR	z-Value	OR
Complaints on chronic pain = yes					2.19	3.57 (1.37–9.30)
Experienced falls = yes					2.70	5.03 (1.88–13.47)
Walking with support or visible impaired = yes			3.89	18.89 (5.45–65.53)	2.83	8.93 (2.50–31.92)

## Data Availability

The data presented in this study are available on request from the corresponding author. The data are not publicly available due to ongoing research.

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
