# Peer review of "Clusters of Physical Frailty and Cognitive Impairment and Their Associated Comorbidities in Older Primary Care Patients"

_healthcare, 2021, doi:10.3390/healthcare9070891_

Round 1

Reviewer 1 Report

Bekic et al. present data on the characteristics of primary care patients with or without frailty and / or cognitive impairment. The idea to use entities defined purely on a functional level and to compare their characteristics based on underlying diseases, comorbidities and risk factors seems interesting. Nevertheless, the research question and accordingly the analyses and the interpretation of the results are not fully clear. The analyses give a bit the impression of entering all the variables that can be found in the electronic health records into the model without having any pre-specified hypotheses or assumptions. Accordingly, the discussion is in some partly lengthy without clear conclusions.

Comments:

Title: I would suggest to add “older”

Introduction:

- The introduction is well written, however the paragraph on “reversible cognitive frailty” is a bit misleading as the authors do not look at modifiable risk factors.

- Aim: In the rest of the manuscript, it is not really clear how the second aim which relates to age is tested.

METHODS:  

- The authors should explain how many primary care facilities participated, how they were selected and how representative the included patients are for the population older than 60.

- I do not understand the preliminary analysis of the incomplete data. Is this a way of data imputation? Does this mean that in 60% of cases variables were missing? Which variables were mostly missing?

- Multinominal logistic regression models: Please explain which variables entered the model. The number of variables listed in table 3 and 4 seem too many when taking into account the low number of people in cluster CF and PhyF. As the authors would like to analyse the effect of age, was age entered in the final model?

RESULTS:

- Table 1 and 2 describe how well the cluster have been defined according to MMSE and the frailty test and how they related to the demographic variables sex and age. Tables 3 and 4 show the comparisons between the clusters and the variables are presented according to the type of variable (numerical or categorial). While this is understandable from a statistical point, this does not help for the interpretation of the results. The variables seem to be at different levels, some are risk factors, some are indicators or consequences for others. Especially, the mixture of laboratory data and diseases is confusing because laboratory data are used for the diagnosis of diseases. For example falls can be consequences of stroke or Parkinson’s disease, related to osteoporosis or related to medications described for another disease. Altogether the factors investigated by the authors may give some indications about the relationships of the underlying processes. However, there are many possible interactions and interpretations. In view of the inflammaging theory, the authors should have attempted some sort of classification or should have formulated some hypotheses before doing the multivariate analyses. The interpretation is otherwise very difficult; this is also reflected by the discussion.

TABLES:

I would suggest to use <0.001 instead of 0.000

Table 1: the pairwise comparisons in the column (post-hoc) are unclear to me. E.g. CI:CF:PHyF does this mean that all of them differ from HF or that CI differs significantly from CF and PhyF, and that CI and PhyF differ significantly?

Table 2: The row “all” after frail can be deleted.

Table 3: What is the meaning of the “*”? If this does mark “means”, why are median and means mixed within the same variable?

Figure:

The authors should use the same y-axis, i.e. 100% for both graphs, otherwise it is difficult to compare cluster 1, 2 and cluster 3,4.

It would be helpful if the clusters are named in addition to the numbers 1-4.

Table 5: The variables should have names.

DISCUSSION:

The authors do their best to explain their results however there is also a lot of speculation.

What is definitively missing and has to be added to the discussion, are the limitations of the study. For example: the low number of persons in PhyF and CF cluster (from a statistical point but also because frail and immobile persons might have be treated in other facilities), the possibility for generalisation of the results, possible bias due to recruitment, confounders / variables that were not available and should have been entered in the analysis, the limitations of the MMSE for detecting mci.

Author Response

Answers to the reviewers` comments

Reviewer 1:

Bekic et al. present data on the characteristics of primary care patients with or without frailty and / or cognitive impairment. The idea to use entities defined purely on a functional level and to compare their characteristics based on underlying diseases, comorbidities and risk factors seems interesting. Nevertheless, the research question and accordingly the analyses and the interpretation of the results are not fully clear. The analyses give a bit the impression of entering all the variables that can be found in the electronic health records into the model without having any pre-specified hypotheses or assumptions. Accordingly, the discussion is in some partly lengthy without clear conclusions.

Comments:

Title: I would suggest to add “older” - done

Introduction:

- The introduction is well written, however the paragraph on “reversible cognitive frailty” is a bit misleading as the authors do not look at modifiable risk factors. – this part was excluded from this section, and the section was a little bit modified

- Aim: In the rest of the manuscript, it is not really clear how the second aim which relates to age is tested. – it is now added in the form of the additional MLR model testing the impact of age and gender to membership to the clusters (Table 6). We have tried to make an adjustment of other MRL models (Tables 7-10) according to age and gender but the produced models were very simple (e.g. with only one variable being contained in the model with numerical variables indicating anthropometric measures and laboratory tests). In our opinion, this phenomenon is a consequence of the fact that both the variable “age” and some other variables are closely associated with membership to the clusters.

METHODS:  

- The authors should explain how many primary care facilities participated, how they were selected and how representative the included patients are for the population older than 60. – these issues are now in a detail elaborated in the Methods section.

- I do not understand the preliminary analysis of the incomplete data. Is this a way of data imputation? Does this mean that in 60% of cases variables were missing? Which variables were mostly missing? – this part of the text was reworded and presented more clearly.

- Multinominal logistic regression models: Please explain which variables entered the model. The number of variables listed in table 3 and 4 seem too many when taking into account the low number of people in cluster CF and PhyF. As the authors would like to analyse the effect of age, was age entered in the final model? – these issues are now clearly explained in the Methods section. All numerical variables, before entering the model, were assessed on collinearity and multilinearity, and variables showing high collinearity and multicollinearity effects were excluded from the input. These variables were: the number of medications prescribed with the effect on mental function (Table 7), and bmi, waist circumference, erythrocyte number, and total cholesterol (Table 8).

For the models indicating diagnoses of chronic diseases and particular functional/sensory organ impairments (Tables 9 and 10), we used only variables that were shown significant in the analysis of differences (because of the large number of these variables and the low number of patients having some of the examined conditions).  

The tables contain only predictors from the generated MLR models with p-value less than 0.05

 RESULTS:

- Table 1 and 2 describe how well the cluster have been defined according to MMSE and the frailty test and how they related to the demographic variables sex and age. Tables 3 and 4 show the comparisons between the clusters and the variables are presented according to the type of variable (numerical or categorial). While this is understandable from a statistical point, this does not help for the interpretation of the results. The variables seem to be at different levels, some are risk factors, some are indicators or consequences for others. Especially, the mixture of laboratory data and diseases is confusing because laboratory data are used for the diagnosis of diseases. For example, falls can be consequences of stroke or Parkinson’s disease, related to osteoporosis or related to medications described for another disease. Altogether the factors investigated by the authors may give some indications about the relationships of the underlying processes. However, there are many possible interactions and interpretations. In view of the inflammaging theory, the authors should have attempted some sort of classification or should have formulated some hypotheses before doing the multivariate analyses. The interpretation is otherwise very difficult; this is also reflected by the discussion. – the old Table 3 is now split into 2 new Tables – where Table 3 indicates the level of comorbidity (variables: total number of diagnoses, total number of prescribed medications, total number of medications with effect on mental functions, and total number of functional/sensory organ impairments), and Table 4 indicates the health-status, described with anthropometric measures and laboratory tests.

Regarding the created MLR models, as it has already been discussed above, we assessed the impact of age and gender in the separate model (Table 6).

We assessed variables indicating the level of comorbidity in the MLR model presented in Table 7 (variables: total number of diagnoses of chronic disease, total number of functional/sensory organ impairments, total number of prescribed medications, and total number of prescribed medications with the effect on mental functions) – the variable “total number of medications prescribed with the effect on mental functions” was excluded from the input because of the collinearity phenomena.

In Table 8, we examined the associations of variables indicating the health-status, described with anthropometric measures and laboratory tests – in the input, there were variables: mid-arm-circumference, fasting glucose, ldl-cholesterol, hdl-cholesterol, creatinine, glomerular filtration rate, C-reactive protein, Haemoglobin, and haematocrit, whereas variables: BMI, waist circumference, total cholesterol, and erythrocyte count, were excluded from the analysis because of the collinearity or multicollinearity.

In Tables 9 and 10, we examined the impact of particular diagnoses of chronic diseases and of functional/sensory organ impairments on membership to the clusters. Only variables that were shown significant in the analysis of differences, entered the model (for reasons that were elaborated above).

TABLES:

I would suggest to use <0.001 instead of 0.000 - done

Table 1: the pairwise comparisons in the column (post-hoc) are unclear to me. E.g. CI:CF:PHyF does this mean that all of them differ from HF or that CI differs significantly from CF and PhyF, and that CI and PhyF differ significantly? – these relationships were presented precisely – please, see the cleared paper!

Table 2: The row “all” after frail can be deleted. - done

Table 3: What is the meaning of the “*”? If this does mark “means”, why are median and means mixed within the same variable? We tested the normality for a particular numerical variable in each cluster. Based on the result, we chose the combination median (IRQ) or mean ± SD.

Figure:

The authors should use the same y-axis, i.e. 100% for both graphs, otherwise it is difficult to compare cluster 1, 2 and cluster 3,4. - done

It would be helpful if the clusters are named in addition to the numbers 1-4. - done

Table 5: The variables should have names. - done

DISCUSSION:

The authors do their best to explain their results however there is also a lot of speculation.

What is definitively missing and has to be added to the discussion, are the limitations of the study. For example: the low number of persons in PhyF and CF cluster (from a statistical point but also because frail and immobile persons might have be treated in other facilities), the possibility for generalisation of the results, possible bias due to recruitment, confounders / variables that were not available and should have been entered in the analysis, the limitations of the MMSE for detecting mci. – the Discussion section was rewritten, to make the comments as much explicit as possible, and to follow the strong logic of data presentation.

According to your advice, we added the limitations of the study paragraph into the discussion section. Thank you for your clarifications and the valuable comments.

Reviewer 2 Report

This is a cross-sectional study, in which the authors identified 4 clusters (HF, CI, CF and PhyF) of physical frailty and cognitive impairment in older primary care patients. The HF cluster showed the younger individuals compared to the rest of the clusters, and gender differences were found just in the PhyF cluster. Then, the authors associated the clusters with the comorbidity patterns. The PhyF cluster has the highest number of chronic disease diagnoses and in consequence, it is the cluster with the highest number of prescribed medications. Cardiometabolic disorders together with the age are a pattern that can distinguish each cluster identified in this study.

Minor comments
1.    Authors should describe all the abbreviations used in the abstract section. 
2.    Figure 1 should be more friendly (review the attached option).
3.    In figure 1 each cluster should be named how was classified and the scale used should be the same in each cluster.
4.    In figure 1, What is the percentage of subjects that present more than 1 chronic disease and functional organ impairment in each cluster (this information should be added in the results).

Author Response

Answers to the reviewers` comments

Reviewer 2:

This is a cross-sectional study, in which the authors identified 4 clusters (HF, CI, CF and PhyF) of physical frailty and cognitive impairment in older primary care patients. The HF cluster showed the younger individuals compared to the rest of the clusters, and gender differences were found just in the PhyF cluster. Then, the authors associated the clusters with the comorbidity patterns. The PhyF cluster has the highest number of chronic disease diagnoses and in consequence, it is the cluster with the highest number of prescribed medications. Cardiometabolic disorders together with the age are a pattern that can distinguish each cluster identified in this study.

Minor comments

  1.    Authors should describe all the abbreviations used in the abstract section. – done (except for the term SD (standard deviation), as it is a widely used abbreviation in the scientific texts.
    2.    Figure 1 should be more friendly (review the attached option). - done
  2.    In figure 1 each cluster should be named how was classified and the scale used should be the same in each cluster. - done
  3.    In figure 1, What is the percentage of subjects that present more than 1 chronic disease and functional organ impairment in each cluster (this information should be added in the results). The proportion of participants with < 3 diagnose vs, those having ≥3 diagnoses of chronic diseases – added to Table 2 (Descriptive statistics – supplementary materials) and to Table 3 (analysis of differences).

The Aknowledgment was added: This work was partially supported by the Slovak Grant Agency of the Ministry of Education and Academy of Science of the Slovak Republic under grant no. 1/0685/21 and The Slovak Research and Development Agency under grants no. APVV-16-0213 and APVV-17-0550.